# Radiation on Earth or in Space: What Does It Change?

**DOI:** 10.3390/ijms22073739

**Published:** 2021-04-03

**Authors:** Juliette Restier-Verlet, Laura El-Nachef, Mélanie L. Ferlazzo, Joëlle Al-Choboq, Adeline Granzotto, Audrey Bouchet, Nicolas Foray

**Affiliations:** Inserm, U1296 Unit, «Radiation: Defense, Health and Environment», Centre Léon-Bérard, 28, Rue Laennec, 69008 Lyon, France; juliette.restier-verlet@inserm.fr (J.R.-V.); laure.el-nachef@inserm.fr (L.E.-N.); melanie.ferlazzo@inserm.fr (M.L.F.); joelle.al-choboq@inserm.fr (J.A.-C.); adeline.granzotto@inserm.fr (A.G.); audrey.bouchet@inserm.fr (A.B.)

**Keywords:** space radiobiology, radiosusceptibility, radiosensitivity, radiodegeneration

## Abstract

After having been an instrument of the Cold War, space exploration has become a major technological, scientific and societal challenge for a number of countries. With new projects to return to the Moon and go to Mars, radiobiologists have been called upon to better assess the risks linked to exposure to radiation emitted from space (IRS), one of the major hazards for astronauts. To this aim, a major task is to identify the specificities of the different sources of IRS that concern astronauts. By considering the probabilities of the impact of IRS against spacecraft shielding, three conclusions can be drawn: (1) The impacts of heavy ions are rare and their contribution to radiation dose may be low during low Earth orbit; (2) secondary particles, including neutrons emitted at low energy from the spacecraft shielding, may be common in deep space and may preferentially target surface tissues such as the eyes and skin; (3) a “bath of radiation” composed of residual rays and fast neutrons inside the spacecraft may present a concern for deep tissues such as bones and the cardiovascular system. Hence, skin melanoma, cataracts, loss of bone mass, and aging of the cardiovascular system are possible, dependent on the dose, dose-rate, and individual factors. This suggests that both radiosusceptibility and radiodegeneration may be concerns related to space exploration. In addition, in the particular case of extreme solar events, radiosensitivity reactions—such as those observed in acute radiation syndrome—may occur and affect blood composition, gastrointestinal and neurologic systems. This review summarizes the specificities of space radiobiology and opens the debate as regards refinements of current radiation protection concepts that will be useful for the better estimation of risks.

## 1. Introduction

To date, exposure to ionizing radiation (IR) is one of the major concerns for space exploration [1,2,3,4]. IR remove electrons from atoms, which triggers the production of reactive oxygen species (ROS) and DNA damage, leading to significant injuries at the molecular, cellular, tissue and/or clinical levels in a dose-dependent manner [5]. While radiobiology is a pluridisciplinary study of the biological effects of IR, space radiobiology is a particular subdomain, in which investigations are made difficult by the very specific physical features of IR emitted from space (IRS). While the sources of IRS are well known, the contribution of each type of IRS to the dose delivered to astronauts needs to be better documented. Furthermore, since space missions generally correspond to exposure to low-dose radiation, space radiobiology should also integrate the uncertainties related to rare physical events, the specificity of some radiobiological phenomena occurring at low-doses, and their contribution to radiation-induced risks [6]. In addition, a complete review of the specific human organs/tissues at risk of low-dose radiation is required, while the specificities of the clinical risks related to cellular death (radiosensitivity), cellular transformation (radiosusceptibility) and cellular aging (radiodegeneration) should be identified and documented. In addition, it is noteworthy that the perception of these clinical consequences by society is different [5,7] (also see below). Lastly, the current international radiation protection recommendations do not integrate individual factors in the radiation response or various specific features of low-energy particles that may be important in the context of space radiobiology [5]. Hence, data controversies, differences in the estimations of the risks and confusion in the terms used have progressively appeared, together with societal interest in the field which has been enhanced by topics such as the International Space Station (ISS) end-of-life and the initiation of projects to travel to the Moon and to Mars. Therefore, better documenting these different issues and clarifying the messages represent the major objectives of this review.

## 2. The Three Sources of Ionizing Radiation Emitted from Space

### 2.1. Historical Features

As part of his work related to low electrical currents, and just before the discovery of radium thanks to Marie Curie, his wife, Pierre Curie developed very sensitive electrometers at the beginning of the 20th century [8]. In 1910, by setting up a Curie electroscope at the top of the Eiffel tower, Wulf, a Jesuit priest, demonstrated that 50% of the electricity in the air was lost by comparison with that assessed at the Earth’s surface [9]. In 1912, by using balloons, Hess demonstrated that the ionization density of the atmosphere decreases up to 1000 m, but increases above 1800 m, suggesting the existence of radiation emitted from space, *the cosmic rays*—even though this term was first proposed by Millikan. Later, with more specific tools, it was shown that cosmic rays are composed of high energy particles. The Nobel Prize in Physics was awarded in 1936 to Hess for his work related to space radiation. In the same year, Solomon published a theory about the interaction between matter and cosmic rays [10]. Between 1930 and 1940, a body of research related to the nucleosynthesis of the universe led to the production of evidence that heavy elements are abundant in the cosmos, namely, *galactic cosmic radiation* (GCR) [6,9,11,12] (Figure 1).

The 1950s correspond to a technological revolution in the field of cosmology: experiments with balloons were replaced by physics and chemistry assessments in artificial satellites. Notably, the data generated by the first Explorer and Pioneer satellites permitted Van Allen and Franck to point out the existence of the Earth’s radiation belt; that is, a high intensity band of corpuscular radiation temporarily trapped in the Earth’s magnetic field, the *Van Allen radiation belt*. They revealed, thereafter, that this belt is composed of protons and electrons that are mainly generated as secondary particles from neutrons emitted from cosmic charged particles [13,14]. In the same period, considerable advances were made in regard to solar radiation. Notably, protons were also considered to be the most abundant particles emitted from the Sun [15,16].

In the 1960s, thanks to the acceleration of spatial technology progress emulated by the Cold War, the specific composition and energy-distribution of the inner and outer Van Allen radiation belt and solar flares were almost completely defined. Astrophysicists and geophysicists pointed out a specific area where the inner Van Allen radiation belt is the closest to the Earth surface, (above Brazil and South Atlantic Ocean): The *South Atlantic Anomaly* (SAA). This area is characterized by an impressive flux of protons and electrons (100–1000 protons/cm^2^/s at 500 km). Numerous hypotheses were formulated about the injuries that such particle flux may cause on electronics, materials, and overall, on human health at each passage above the SAA. From this period, on-board radiation dosimetry was systematically set up for each space mission [16,17,18].

### 2.2. From the Cosmos to the Shielding: The Three Sources of IRS and Their Physical Features

To date, IRS is generally divided into three major sources: the cosmos, the Sun and the Van Allen radiation belt. Their physical features are fairly well known [16,17] (Figure 1):
-*The cosmos: Galactic cosmic radiation* (GCR) is composed of 85% protons, 14% helium ions, and about 1% heavier elements, such as iron. In GCR, the corresponding fluxes are about four particles/cm^2^/s for protons, 0.4 for helium ions and 10^−4^ to 10^−2^ for heavier ions. Their energy can be very high (more than 10^11^ MeV). The probability of impact of the heaviest ions to astronauts can be considered as negligible in low Earth orbit (LEO) but not in deep space [19]. For quantifying the risks during LEO, GCR can be reasonably reduced as a flux of protons and helium ions [6].-*The Sun:* Solar flares generally consist of 92% protons, 6% helium ions and about 2% heavier elements at various MeV values. The solar wind is an intense (about 10^8^ protons/cm^2^/s) flux of protons at various keV values. Solar flares, storms and winds are infrequent events (generally obeying a cycle of 11 years for solar flares) [20]. Even if space missions are scheduled to avoid such events, some missions, such as Apollo-XIV, Skylab-4, and Mir-15, have been exposed to some significant solar events, at least partially: The crews received 11.4 mGy in 9 days, 77 mGy in 90 days, and 92.9 mGy in 185 days. All these values were obtained from on-board dosimeters that included the GCR contribution [6] (Figure 2). However, no significant injury was reported to astronauts or electronics, probably because the duration of the mission was short and/or the excess of dose remained limited. Predicting solar events should, therefore, be an essential part of ensuring the radiological protection of astronauts. By excluding exceptional solar events, one can reasonably consider the Sun as a simple source of protons emitted at high flux but at a relatively low energy.-*The Van Allen radiation belt* is made of two layers: An “outer” belt composed of electrons, the energy of which ranges from 40 keV to 7 MeV (in electron radiotherapy, the energy of electrons is 6–25 MeV), and an “inner” belt, mostly composed of protons, the energy of which ranges from 100 keV and 400 MeV (in proton therapy, the energy of protons is 60–280 MeV). As described above, the particularity of the Van Allen belt is the excess of radiation observed in the SAA. At each passage above the SAA, electronic devices and astronauts receive a peak of dose that may represent a dose six-fold higher than the average one experienced outside the SAA [18]. Lastly, it is noteworthy that inside the Van Allen belt, the flux and the nature of the particles that impact the spacecraft are greatly conditioned by the flight parameters (notably, day/night and orbit inclination [21,22]. Hence, significant differences may appear when comparing data from the Apollo missions—that spent some time outside the Van Allen belt—and the ISS that remains in LEO, protected inside the magnetosphere [23]. The assessed and calculated doses are discussed below.


By integrating all these data and excluding exceptional solar events, an empirical formula can be deduced between the flux, *F_outside_*—assessed outside the spacecraft and expressed in particles/cm^2^/s—at the energy, E, expressed in MeV. This formula fits the proton component particularly well:(1)$$F_{outside}\left(E\right)={\left(\frac{10}{\sqrt{E}}\right)}^{5}$$

### 2.3. Inside the Shielding: Occurrence of a Variety of Electronic and Nuclear Reactions

Since the 1960s, the shielding composition designed to protect astronauts and electronic equipment has been the subject of a number of reports [2,17,24,25,26,27,28,29,30]. While very dense elements such as lead or depleted uranium would theoretically offer the best protection against radiation, the aluminum-containing shieldings have been considered as the best compromise between the highest density and the best protection. However, to date, there are a plethora of projects focused on different new shieldings combining multilayers of dense elements that stop charged particles and materials of low atomic numbers, such as hydrogen, that absorb neutrons [2,17,24,25,26,27,28,29].

However, whatever the final shielding, the energy and flux ranges of IRS are impossible to reproduce on Earth. Consequently, the radiation dose received by astronauts inside their spacecraft must be calculated from the physical interactions between the IRS and the shielding components, simulated by a Monte Carlo approach and transport equations. This task is particularly difficult since the choice of the simulation code, the values of the cut-off energies and the uncertainties of cross-sections related to reactions involving high-energy particles, represent sources of discrepancies between research groups [31].

The shielding serves as a *filter* and is a *transformer*: In other words, the collision between the IRS and the shielding provides primary and secondary rays and particles that obey the following trends.

-*A general decrease in the flux:* The flux of the secondary particles provided from the spacecraft shielding must be lower than that of the incident particles. A decreased flux leads to a lower absorbed dose: By omitting exceptional solar events, the dose delivered to astronauts is generally in the order of mGy per day. The decrease in the flux by the shielding can be roughly represented by a vertical shift in the data plotted in Figure 1.-*A general degradation of the energy combined with changes in the nature of radiation*: The energy of the secondary particles provided from the spacecraft shielding must be lower than the incident particles and their nature can be changed. The decrease in energy can be roughly represented by a horizontal shift in the data plotted in Figure 1. High-energy particles interact with matter to produce atomic displacements and/or electronic ionizations and excitations. If the kinetic energy of the incident particle transferred to the nucleus of the atoms of the shielding is sufficient, the moving atom may serve as a projectile to produce secondary displacements or ionize or excite other atoms adjacent to its path [32]. For example, while electrons emitted from space may be stopped by shielding, they can produce a non-negligible effect through the bremsstrahlung phenomenon, which results in an intense ray build-up behind the shielding [21,22]. With regard to protons emitted from space, numerous excitations and ionizations are expected inside the shielding, together with the emission of low-energy metal ions (LEMI) [32,33]. In addition, secondary neutrons provided from the aluminum-containing shielding have also been detected in low earth orbiting spacecraft [34,35]. However, although their energy spectrum is difficult to measure, the long duration exposure facility (LDEF) mission has provided data suggesting that neutrons of more than 1 MeV are the most abundant and responsible for most of the total equivalent neutron dose [36]. At this stage, it is important to address whether neutrons may be responsible for some of the activation reactions in the spacecraft. In general, the mechanical properties of metals and ceramics that are present in the spacecraft do not significantly suffer from a flux that is lower than 10^17^/cm^2^ for protons > 1 MeV, 10^17^/cm^2^ for neutrons > 1 keV and 10^18^/cm^2^ for electrons > 1 MeV [32]. The neutron activation and the neutron photo-emission or spallation of oxygen inside the spacecraft may result in ^19^O and ^15^O, respectively. However, the lifetimes of these oxygen isotopes are too short to obtain reliable measurements. Conversely, ^7^Be, ^22^Na and ^24^Na, are the major activable gamma-ray-emitting radioisotopes present in the body, with half-lives that are long enough to be measurable. Additionally, in 1969, Brodzinski et al. proposed to quantify the radiation dose to Apollo astronauts from the assessment of such activated nuclides. Relevant dosimetry data were obtained, suggesting that measurable activation events occur in the spacecraft [37]. Unfortunately, to our knowledge, no data have been published in regard to this technique being applied in the ISS, maybe because the physical conditions of neutron activation are not so favorable at LEO. Further investigations are, however, needed to document the impact of this particular process on the dose.

### 2.4. From the Shielding to Astronauts: The Contribution of Each IRF to the Radiation Dose

From the two trends explained above, three major types of radiation must be considered: *Low-energy particles* (mostly protons, electrons and metal ions directly produced by the shielding), neutrons, and a “*bath of radiation*” resulting from a mixture of all the secondary rays that result from Bremsstrahlung and activation emitted in the spacecraft. To illustrate these specific types of rays and particles emitted inside the spacecraft, presented below are two representative examples:-At the end of the 1960s, it was reported that helmets used during the Apollo missions were impacted by specific low-energy metal ions (LEMI). As described above, the probability of the impact of such GCR metal ions is approximately a few particles per km^2^ per century. Thus, the LEMIs detected in the helmets did not come from the cosmos, but were generated from the walls of the spacecraft made of iron, zinc, aluminum, nickel, and copper. These particles resulted from the interaction between the incident protons from space with the metal shielding. LEMIs were found to emit at a few keV to several MeV [33]. In addition to these LEMIs, some low-energy protons (LEP) and low-energy electrons (LEE) can be emitted from the shielding and may have a significant impact on electronics [38]. It is noteworthy that LEMI, LEP and LEE deliver nearly all their energy at the surface of matter (with an average track path of several mm to nm) [39,40]. Hence, these particles preferentially target the external parts of the human body, such as the skin and eyes (Figure 1). It is noteworthy that this low-energy particle component may be reduced in the ISS in comparison with deep space missions.-The other rays and particles directly or indirectly produced by these secondary particles represent a constant “bath of radiation”, notably made of the build-up of X-rays and gamma-rays. An overview of the dosimetry of all space missions, from that of Gagarin in 1962 to the most recent with the Space Shuttle, has shown that the dose received by astronauts is strongly time-dependent [6] (Figure 2). Whether the mission concerns LEO, the surface of the Moon or a trip from Earth to the Moon (Apollo missions only), the data review indicates a constant dose rate of about 0.4 mGy/d (Table 1). Such a “bath of radiation” is mainly made of X-rays and gamma-rays and should concern the human body as a whole, even at depth [6].

Despite the relative simplicity of the above description, the contribution of GCR and that of neutrons to the dose has been a subject of controversy. For example, in 2001, Benton and Benton (2001) reported that “roughly half the dose on the ISS is expected to come from trapped protons and half from GCR” [41], while Cucinotta et al. (2008) supported that “80% or more of organ dose equivalents on the ISS are from GCR and only a small contribution from trapped protons” [42]. As described above, such differences are likely due to different simulation methodologies and reveal the difficulties of measuring and predicting a very complex spectrum of rays and particles emitted from space and targeting the spacecraft. These differences could also be due to the meaning of the term “GCR” that may evoke connotations of heavy ions only and not necessarily protons, helium ions and heavier ions.

Similarly, the contribution of neutrons to the dose is still not well established, notably because of the difficulties to introduce high-energy neutrons features into measurements and calculations. While the contribution of neutrons with an energy lower than 1 MeV to the dose has been found to be less than 5%, calculations show that 1–10 MeV neutrons should contribute to half of the total dose in the space shuttle [34,43]. More recently, the measured neutron spectrum (12–436 MeV) in the ISS was found to produce a dose rate of 3.8 ± 1.2 μGy/d. Extrapolation of the spectrum at 0.1–1000 MeV led to a total neutron-induced dose rate of 6 ± 2 μGy/day [44].

In addition to these uncertainties regarding high-energy neutrons, the estimation of the dose is made difficult by the dependence of GCR fluxes with the solar cycle (*solar modulation*): The GCR flux is maximal when solar activity is minimal. Differences between solar minimum and solar maximum are a factor of approximately five and also explain the range of uncertainties [19,29]. Similar explanations are relevant for explaining the night/day differences on the Moon and Mars surface [45].

### 2.5. From the Assessment of the Absorbed Dose to the Calculation of the Effective Dose

The same absorbed dose does not lead to the same biological effects, rather it is dependent on the type of radiation and/or the irradiated tissue. For this reason, the *equivalent dose* is a useful way to calculate the radiation-induced risk when exposed to IR involving different types of radiation. Similarly, the effective dose permits one to quantify the radiation-induced risks when different parts of the body, sensitive or not, are exposed to IR. As part of the international radiation protection recommendations of the International Commission of Radiological Protection (ICRP), calculation of the equivalent dose H_T_ and effective dose E are based on the dimensionless weighting factors *W_R_* and *W_T_*, related to the nature of the radiation type and the tissue, respectively [46]. The equivalent dose H is defined as the probability of a stochastic event (namely radiation-induced cancer) for an absorbed dose D in the tissue T irradiated with the radiation type R:(2)$$H_{T}=\sum_{R}W_{R}.D_{T,R}$$

The effective dose E is defined as the probability of radiation-induced cancer for a mass-averaged absorbed dose $\bar{D}$ in tissue T, irradiated with radiation type R:(3)$$E_{T}=\sum_{T}W_{T}.H_{T}=\sum_{T}W_{T}.\sum_{R}W_{R}{\bar{D}}_{T,R}$$

The relevance of the dose equivalent and the effective dose depend on the relevance of the weighting factors *W_R_* and *W_T_*. The specificity of space radiobiology is that the *W_R_* of some very high- or very low-energy particles may not be consensual. The relevance of weighting factors W will be discussed in the following paragraphs. Notwithstanding the question of relevance, a non-exhaustive list of reports on estimated doses of spatial interest shows a relatively good agreement between the absorbed doses and the effective doses proposed for the ISS, Moon and Mars [47,48,49,50,51,52] (Table 1). Again, the range of values are explained by solar modulation.

### 2.6. Comparison with the Natural Radiation Backgrounds on Earth, the Moon and Mars

As far as any shielding is used during the exposure, the low-energy particle component does not exist on Earth. In contrast, the natural radiation background can be considered as a “bath of radiation” composed of three major components:-*The telluric radiation component:* The three major radioactive decay chains of natural radiation, such as uranium (^235^U or ^238^U) and thorium (^232^Th), are present on Earth and continuously provide several unstable and stable radionuclides. The major part of the telluric radiation component comes from radon, a naturally radioactive gas resulting from the decay of uranium and radium naturally present in the soil. The average effective dose inhaled in air produced by radon is about 1.3 mSv/y, while terrestrial radiation from the ground represents about 0.5 mSv/y.-*The organic radiation component:* An effective dose-rate of about 0.3 mSv/y comes from organic products that contain naturally radioactive substances such as potassium (^40^K) and carbon (^14^C).-*The cosmic radiation component:* At sea level, an effective dose-rate of 0.3 mSv/y comes from cosmic radiation.

The average worldwide effective dose-rate provided by the natural radiation background is about 2.4 mSv/y. However, this value can vary greatly in various regions of the world. In Japan, it is about 0.5 mSv/y (the lowest value), while in certain regions of Brazil, India, and Iran, the amount of radiation is about 140 times higher and can reach 70 mSv/y (e.g., Ramsar, Iran). Some effective dose-rate values of 260 mSv/y have been also reported in a district of Ramsar [50]. It is noteworthy that the cancer incidence ratio between Ramsar and Japan is likely to be much lower than 140-fold, notwithstanding the confounding factors linked to the environment, suggesting that no evident risk threshold can be pointed out between 0.5 and 70 mSv/y. In contrast, some reports have suggested a lower incidence of cancer and radiation-induced diseases in Ramsar (hormesis phenomenon). However, the number of inhabitants in Ramsar is small. Consequently, further investigations are needed with larger cohorts of individuals [50].

The continuous “bath” of space radiation of about 0.4 mGy/d described above, corresponds to an effective dose rate ranging from 0.1 to 0.3 mSv/d without solar flares and about 0.5 mSv/d with solar flares (Table 1). Considering the average values given in Table 1, one year in the ISS corresponds to an effective dose ranging from 110 to 180 mSv/y. These values are much higher than the average worldwide background. The surfaces of the Moon and Mars appear to be more radioactive: According to the time of day, the natural radiation backgrounds assessed on the Moon and Mars are 110–380 and 130–260 mSv/y, respectively [47,48,49,50,51] (Table 1 and Table 2). This is mainly due to the absence of significant magnetic protection and to a higher telluric component. These values should, therefore, be taken into account in the evaluation of the radiation-induced risks for astronauts (Figure 2) (Table 1). To date, the exploration of Mars is thought to require about 900 days or more with more than one year being spent in deep space, where shielding will not be able to protect astronauts significantly from GCR. The latest estimation of the effective dose for such a mission stated a value of approximately 1200 mSv, corresponding, therefore, to an average of 1.3 mSv/d (i.e., 475 mSv/y, corresponding to 200 times higher than the average worldwide natural radiation background) (Table 1 and Table 2).

Such estimation raises the question of the acceptability of such excess of risk vis-à-vis society and current regulations on Earth. As part of the international radiation protection rules, the annual occupational exposure limit is 20 mSv/y (and no more than 100 mSv in 5 years). This suggests that the limit may be exceeded by space missions of more than 50 days or alternatively, astronauts may partake in a space mission of 250 days but should not take part in any other mission for the next 4 years. The exploration of Mars is far beyond this in terms of the involved risk. Therefore, a prominent question is how to manage such risks?

## 3. The Potential Radiation-Induced Risks for the Astronauts: What Do We Expect?

As part of the international radiation protection rules, radiation-induced risks are generally based on the notion of deterministic/stochastic events and the calculation of the dose equivalent and effective dose [46]. However, as detailed in the next chapter, such approach is not necessarily consensual, notably, to identify all the clinical consequences of an exposure to IR. Therefore, let us consider that IR can produce three distinct types of clinical effects: Radiosensitivity, radiosusceptibility and radiodegeneration [5].

### 3.1. Radiosensitivity in Space

R**adiosensitivity** has been defined as the proneness to adverse tissue effects (burns, dermatitis, proctites), generally attributable to cell death [5]. Radiosensitivity reactions are likely to occur at high doses, often above 0.7 Gy. Radiosensitivity reactions are generally observed during/after radiotherapy treatment due to effects on a specific organ targeted by irradiation, such as dermatitis or proctites or the entire body (notably in case of total body irradiation). The radiopathology of acute radiation syndrome (ARS) is fairly well described: It can occur at 0.7 Gy or more and is generally divided into three sub-syndromes: Bone marrow (0.7–6 Gy), gastrointestinal (6–8 Gy) and neurovascular (8–12 Gy) (Table 2) [53]. So far, no case of ARS has been observed among the astronauts. Except for intense solar events, no radiosensitivity reactions are expected during space missions. The reference to ARS in relation to solar events may explain why blood forming organs, gastrointestinal and central neurologic systems are cited in the risks related to the exploration of Mars [54,55]

With regard to genetic predisposition, a number of syndromes are associated with individual radiosensitivity and have been well characterized to date. However, the clinical features of these syndromes are so precocious and severe that it is unlikely that over-selected astronauts will be affected by such genetic diseases [5]. In conclusion, if radiosensitivity reactions may be theoretically possible during exceptional solar events, none have been observed, likely because all the past space missions have corresponded to too low-doses and no significant solar events. A mission to Mars may raise this question.

### 3.2. Radiosusceptibility in Space

Radiosusceptibility has been defined as the proneness to radiation-induced cancers, generally attributable to cell transformation [5]. Radiosusceptibility reactions are generally encountered either in areas irradiated by radiotherapy (secondary cancers), or after accidental or occupational exposure. The epidemiological data of the Hiroshima bomb survivors represent the only source of consensual data to describe the risk of radiation-induced cancer as a function of dose: The risk of radiation-induced leukemia and solid cancers increases significantly from a threshold dose of 100 and 200 mSv, respectively [56] (Table 2). However, these data are only valid for exposures delivered in a few minutes (flash exposures), which is possible in case of solar events, but does not answer the question of protracted exposures during long space missions. There are no consensual data of radiation-induced cancer as a function of the dose-rate. However, it is considered that as the length of time that the dose is delivered for increases, the less hazardous the radiation response is expected to be [5]. Thus, the threshold dose of radiation-induced cancers for low-dose-rated exposures is expected to be much greater than 100–200 mSv/y, but its actual value needs to be documented. Furthermore, to date, the number of astronauts who have performed long missions of more than 250 days is too low to deduce any reliable numerical value [6]. Despite this lack of data, there are a number of emerging mathematical models that propose cancer risk predictions with different scenarios of space missions [57,58]. To determine the risk of cancer as a function of dose-rate is an important challenge for space radiobiology but also for radiobiology in general. Indeed, the potential cancer risks linked to mammography (about 2 mGy per s) and to CT scan exams (some tens mGy per min) raise the question of the dose-rate effect on cancer occurrence [59,60] (Table 2).

What type of radiation-induced cancer do we expect from exposure to space radiation? Since LEMI, LEP and LEE preferentially target the surface of body, they might increase the occurrence of radiation-induced cancer in the skin and eyes, notably, skin and eye melanoma (Table 3):-Recently, an increase in mortality from skin melanoma has been reported among astronauts. This increase was shown to be consistent with skin melanoma observed among aircraft pilots. Ultraviolet radiation and lifestyle were suggested as potential causes. However, it must be stressed that neutrons, LEMI, LEP and LEE may also present a cause of concern for aircraft pilots, similarly to astronauts. Conversely, ultraviolet radiation cannot be suggested as a cause in the case of astronauts [61,62]. Further investigations are, therefore, needed to document the potential physical causes of skin melanoma in both aircraft and spacecraft crews. It is noteworthy that the physical parameters of the flights (notably LEO or non-LEO) significantly affects the occurrence of such tumors [63].-With regard to radiation-induced eye tumors, although rare, choroid melanoma is the most frequent primarily adult tumor in the eye [64]. However, to our knowledge, no IR-induced choroid melanoma has been observed so far. No choroid melanoma has been reported in astronauts. Retinoblastoma, another common tumor of the ocular system can be evoked. However, the retina may be too deep in the eye to be reached by energy deposition from LEMI, LEP and LEE (the average distance between the iris and retina is about 2 cm). Again, the flight parameters and the relatively low duration of previous space missions may limit the occurrence of such specific events.

Unlike LEMI, LEP and LEE, the “bath of radiation” composed of more energetic rays and particles targets deeper tissues. In this specific case, leukemia represents the most probable radiation-induced cancer. However, Hiroshima bomb survivors’ data suggest that the occurrence of this form of cancer requires much more than a cumulative exposure of 100 mSv, and therefore, much more than 1 year in space. Hence, no radiation-induced leukemia has been reported yet in the astronauts’ corps, probably because previous space missions were not long enough. For the exploration of Mars, such risk has to be considered.

With regard to the potential genetic predisposition that may be a cause for concern among astronauts, a number of genetic syndromes are associated with individual cancer proneness that also increase the risk of radiation-induced cancer. Unlike the syndromes associated with radiosensitivity, the syndromes associated with cancer proneness and radiosusceptibility are generally caused by heterozygous gene mutations and their related clinical features may not be detectable without DNA sequencing before cancer occurrence. This is notably the case of the heterozygous mutations of ATM, BRCA1, BRCA2, p53, and Rb genes that confer a high risk of breast cancer and retinoblastoma, respectively [5]. Additionally, it is noteworthy that carriers of heterozygous mutations of ATM may represent 1% of the whole population [5]. However, there is no known gene that has been identified as being associated with radiation-induced skin or eye melanoma or specific to radiation-induced leukemia. Hence, while some actual risk of radiation-induced melanoma emerges, further investigations about specific biomarkers of radiosusceptibility are needed (Table 3).

### 3.3. Radiodegeneration in Space

Radiodegeneration has been defined as the proneness to radiation-induced accelerated aging of specific tissues, generally attributable to tolerance of a certain amount of DNA damage [5]. The most frequent radiodegeneration reactions include radiation-induced cataracts. To date, this appears as the most probable consequence of exposure to cosmic radiation [65]. Forty-eight cases of severe lens opacification (16.2%) were observed among the 295 NASA astronauts who participated in the LSAH (Longitudinal Study of Astronaut Health) study, but 86% of the astronauts who stayed in space suffered from a pathology of the eye. An increased risk of cataracts was generally observed after exposure to doses above 8 mGy, which corresponds to approximately 20 mission days [66,67]. Interestingly, on Earth, radiation-induced cataracts have long been considered as a relatively rare condition, requiring an estimated dose threshold of about 2 Gy to the eyes. However, recent data suggest that this dose threshold could be much lower than 2 Gy. A general discussion about the relevance of the ICRP recommendations related to the response of the ocular system to IR can be found in the next chapter [6].

In addition to the acceleration of aging of lens that leads to cataracts, the potential radiation-induced aging of the skin is another pressing concern. Surprisingly, while the skin represents a much greater surface area for the impact of LEMI, LEP and LEE than lens, to date, no quantitative assessment of space radiation-induced aging of the skin has been conducted. Such an absence of any data may be explained not only by the current wearing of worksuits during space missions (only the forearms may be eventually exposed inside the spacecraft), but also by the lack of specific biomarkers for aging [5]. The location of skin melanoma or any change of texture should be therefore registered carefully. The question of radiodegeneration has been recently raised by space experiments on twins [68,69,70,71,72]. During a one-year ISS mission, one male twin was on board while his monozygotic twin served as a genetically matched ground control. His telomeres were found to be longer during spaceflight but shortened rapidly upon return to Earth. An increase in the number of chromosomal inversions was also found to persist after spaceflight [69]. However, while telomere biomarkers have been found to be associated with aging, further investigations are needed to demonstrate whether chromosomal inversions are correlated with radiosusceptibility or radiodegeneration [5].

With regard to the potential aging consequences of the “bath of radiation”, two specific tissues should attract our attention: Bone and the cardiovascular system (Table 3):-The loss of bone mass in astronauts, especially in weight-bearing bones, is a current observation performed after each space mission, that may have consequences for the immune system. The loss of bone mass was hypothesized to be similar to osteoporosis [73,74,75,76]. Although the inherent mechanisms of this loss of bone mass have been based on animal models data, it appears that a reduced osteoblast function leads to decreased bone formation, while bone resorption is unaltered or increased. The loss of bone mass has long been attributed to microgravity [73]. However, while the molecular and cellular pathways by which microgravity may act in the form of biochemical signals are still unknown, some emerging data suggest that radiation (and logically, the “bath of radiation”) may also affect bones [77]. Interestingly, by radiobiologically characterizing various genetic syndromes associated with facial dysmorphy, osteoblasts were shown to be more radiosensitive than the skin of the same donor, which may suggest that IR contribution to the loss of bone mass in astronauts has been underestimated [78]. Further studies are, therefore, needed to document the radiodegeneration of bone in response to IRS. In the following chapters, we discuss the advantages of anti-osteoporosis drugs as countermeasures.-Epidemiology data of women patients with breast cancer have shown that more than 50% of women are at risk of a heart attack for 10 years post-radiotherapy [79,80,81,82]. Such examples, suggest that the cardiovascular system may be mechanically affected by radiation and sensitive to low-dose. To date, there is no evidence of any cardiovascular disease in astronauts caused by exposure to IRS [62,83]. Again, the microgravity contribution should be separated from the radiation contribution [84]. However, even though the cohorts investigated have been too small, the potential radiodegeneration of heart tissue needs to be further investigated.

## 4. Radiation Protection Factors and Radiobiological Effects Specific to Low-Dose Radiation to Be Considered

### 4.1. Radiation Protection Factors to Be Investigated to Refine the Estimation of Risks

While the calculations of the risks are based on the dose equivalent and effective dose defined by the formulas (2) and (3), it is noteworthy that the weighting factor *W_R_* has been defined as equal to 1 for electrons, X-rays and gamma-rays, and equal to 2 for protons regardless of their energy levels [46]. However, there is now evidence that the considerable energy deposed locally by LEMI, LEP or LEE at the surface of the matter has more significant biological and clinical consequences than the same particles being emitted at a higher energy [85,86]. Hence, the definition of the *W_R_* for electrons and protons should be refined and become a function of energy.

In addition to the problem of the energy-dependence of *W_R_*, it must be stressed that, a given *W_T_* corresponds to a mis-defined clinical consequence. For example, when the *W_T_* of the eyes, breast and skin is considered, it is more likely to illustrate the risk of cataracts, radiation-induced breast cancer and the occurrence of radiodermatites, respectively [87]. Hence, to date, it is necessary to evaluate the risks of radiosensitivity, radiosusceptibility and radiodegeneration, separately, since the management, cure and prevention of radiation-induced cataracts, breast cancer and dermatitis are not the same.

Lastly, even if the individual proneness to radiosensitivity, radiosusceptibility and radiodegeneration is limited by the over-selection of astronauts, it is practically possible that an individual who appears to be apparently healthy up to the age of 40 could be a carrier of a cancer-prone gene mutation. Let us return to the fact that ATM heterozygotes represent about 1% of the whole population. Hence, it is important to consider the excess of risks due to the genetic status of an astronaut, that may be more than 10 according to the gene mutations [5]. Consequently, a new weighting factor quantifying the individual factors involved may be useful.

Taken together, our examples suggest that space radiobiology raises important questions related to the ICRP recommendations and the definition of some major notions of radiation protection.

### 4.2. Radiobiological Effects Specific to Low-Dose Radiation to Be Investigated to Refine the Estimation of Risks

In addition to the refinements proposed above, it is necessary to discuss some specific radiobiological phenomena that render the dose–response curve non-linearly dose-dependance and the quantification of the risks more complex.

*The hypersensitivity to low-dose (HRS) phenomenon* was described for the first time by Lambin et al. (1993) [88] and Marples and Joiner (1993) [89]. In vitro, this phenomenon results in a significant reduction in clonogenic cell survival, an increase in chromosome breaks, micronuclei, unrepaired DNA double-strand breaks (DSB) and/or gene mutations after a single low-dose generally belonging to the 100–800 mGy dose range. However, this range may vary with the dose-rate and the HRS may be maximal at lower doses [90]. At a dose-rate higher than Gy/min, the maximal HRS effect is generally obtained at 200 mGy in human cells and corresponds to a biological effect equivalent to a dose that is five to ten times higher (i.e., to 1 to 2 Gy) [90,91]. It is noteworthy that this phenomenon has been observed in normal human tissues and, notably, skin cells [92] and a mechanistic model has been recently proposed [93,94]. While no current radiation protection regulation integrates this yet, the HRS phenomenon may theoretically increase the risks related to small solar events (between 100 and 800 mGy). Further investigations as regards the occurrence of this phenomenon are needed.

*The hormesis phenomenon* was described for the first time in the radiation research field by Luckey [95]. It is described as a J-shaped dose- or dose-rate dependent phenomenon, associated with a specific threshold, under which, stress is considered to be “positive” and above which it is detrimental [96,97]. Hormesis appeared to be more frequently observed in human untransformed radioresistant cells exposed at the doses belonging to the 20–75 mGy range and corresponds to a negative cancer risk in the Hiroshima survivors data [94]. Again, the occurrence of the hormesis phenomenon depends on the dose-rate but this field needs to be better documented for human cells.

*The bystander effect* is defined as any biological effect expressed in cells that are not directly targeted by irradiation but that are situated within the close vicinity of irradiated cells [98,99]. *Stricto sensu*, the bystander effect is not a radiobiological phenomenon specific to low-dose radiation, but bystander cells generally show levels of DNA and chromosome damage, mutations and cell death very similar to those encountered at low-dose radiation. If a bystander effect occurs after the impact of low-energy particles, the energy deposition that initially targets a small number of cells may indirectly concern a larger surface as if surrounding cells have been exposed to low-dose radiation. Hence, the bystander effect produced by IRS should be documented to refine our estimation of the risks.

## 5. Countermeasures

From the above analyses, we distinguished that low-energy ions/particles that produce a massive energy deposition at the surface of the skin and eyes, and the “bath of radiation” that may impact deeper tissues and neutrons may concern both. The contribution to the total dose of these three types of events can vary with the flight parameters (e.g., LEO vs. deep space). To reduce the specific risks related to low-energy particles, a simple form of protection may be sufficient: Indeed, special glasses and worksuits may protect both the skin and eyes significantly. Despite the fact that the majority of studies concern either extravehicular mobility [100,101] or orthostatic intolerance [102], a water-filled garment to protect astronauts has been developed recently in the ISS [103] (Table 3).

With regard to protection against the “bath of radiation” that may affect deep tissues, considerable efforts have been aimed at developing chemical radioprotectors to reduce the results of radiolysis and the amount of radiation-induced DNA damage. At present, the only effective radioprotective agent to protect healthy tissues during radiotherapy is ethyol (amifostine). Amifostine treatment results in a significant decrease in the amount of DNA damage due to its powerful anti-oxidative effect [104,105,106]. However, this drug induces disabling hypotension and would, therefore, be unusable for the protection of astronauts.

In fact, the great majority of chemical radioprotectors consist of reducing the amount of DNA damage *induced* but not necessarily the amount of DNA damage *to be repaired.* More recently, our lab has proposed a general mechanistic model of radiation action based on the radiation-induced nucleo-shuttling of the ATM protein (RIANS) [107]. After irradiation, the ATM dimers become monomers and diffuse into the nucleus to recognize DNA double-strand breaks and trigger their repair through the phosphorylation of the histone variant H2AX (γH2AX). Any delay in the RIANS causes radiosensitivity, radiosusceptibility and/or radiodegeneration [5]. Interestingly, we have provided clues that a combination of statins (anti-cholesterol-lowering drugs) and bisphosphonates (anti-osteoporosis) was shown to facilitate the entry of the ATM kinase into the nucleus to trigger DNA damage repair [5,78]. Such a treatment (notably combining pravastatin and zoledronate (ZOPRA)) was found to be more effective than current anti-oxidative drugs such as ethyol or N-acetylcysteine (N. Foray, personal communication; paper in preparation). Hence, it may be interesting to test the long-term action of ZOPRA during long-flight missions.

## 6. Conclusions and Perspectives

A careful analysis of the flux/energy-spectrum of space radiation and the study of the secondary particles emitted from the shielding has refined the definition of the potential clinical consequences of occupational exposure to IRS for astronauts. We have identified the risks related to high localized energy deposition due to low energy particles/ions that concern the *surface tissues* (such as the skin and eyes), while the low-dose-rated “bath of radiation”—including fast neutrons—concerns the *deep tissues* (such as the bones and cardiovascular system). By omitting the consequences of severe solar events, both radiosusceptibility and radiodegeneration reactions are expected: Radiation-induced melanomas and cataracts for surface tissues and loss of bone mass and accelerated aging of the cardiovascular system for deep tissues. Specific countermeasures may be considered for each situation: Glasses and special suits for surface tissue risks and chemical radioprotective drugs for the deeper tissue risks. Therefore, the countermeasures represent a major axis of space radiation research. Lastly, a debate about the impact of the specific radiobiological phenomena occurring at low-dose radiation, including the bystander effect and a refinement of the radiation protection rules, should be organized rapidly in collaboration with all the key players in space radiobiology.

## Figures and Tables

**Figure 1 ijms-22-03739-f001:**
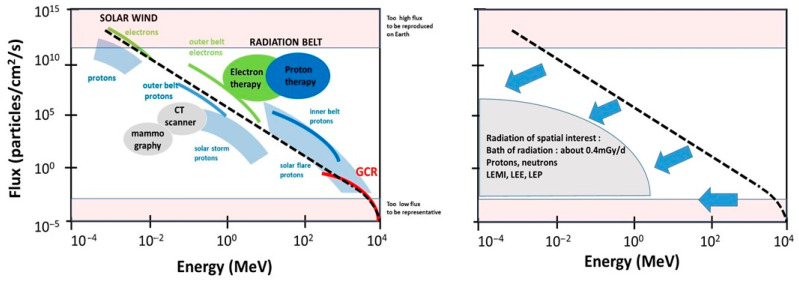
Schematic representation of the 3 major sources of space radiation as a function of energy and flux: Galactic cosmic radiation (GCR), solar radiation component and Van Allen radiation belt. Left panel. The dotted line represents the general relationship between flux and energy. The confidence zones represent the physical features of rays and particles used in proton therapy, electron therapy, computerized tomography (CT) scan and mammography. Right panel. Through their interaction with the shielding, space radiation is degraded in nature, flux and energy to give protons, neutrons, low-energy-metal ions (LEMI), low-energy-electrons (LEE), low-energy-protons (LEP) and a “bath” of residual radiation at about 0.4 mGy/d [6]. Blue arrows represent the effect of the spacecraft shielding to the IRS.

**Figure 2 ijms-22-03739-f002:**
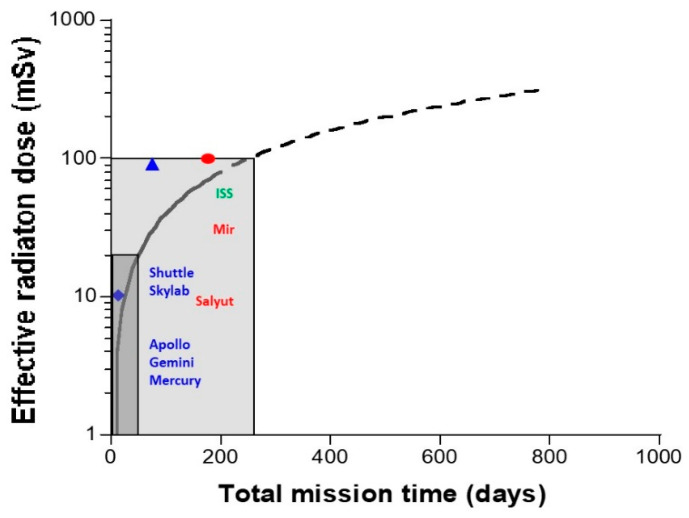
Schematic representation of effective dose per day of space mission. *Left panel.* The black solid line and its dotted line extension represents the average over the whole duration of the mission from all of the space mission’s on-board dosimetry data until 1990s [6]. Blue losange, blue triangle and red circle represent the Apollo XIV, Skylab4 and Mir 15 missions’ on-board dosimetry data, respectively. Dark and light grey zones represent the annual and the 5-year limits of occupational exposure to ionizing radiation (IR) (20 mSv/y and 100 mSv/y, respectively).

**Table 1 ijms-22-03739-t001:** Non-exhaustive list of reports about estimated doses of spatial interest.

Missions	References	Absorbed Dose/Day (mGy/d)	Effective Dose/Day (mSv/d)
Missions before ISS	[41]	Salyut: 0.1–0.3 Apollo: 0.22–1.27	
Missions from Gagarin to Space Shuttle	[6]	0.2–1.26 Average: 0.4	
ISS	[23]		0.153–0.231
ISS during Solar event	[23]		0.535
ISS	[35]		0.15
ISS during Solar event	[20]	10.48	
Moon surface	[19]		0.37–0.97
	[47]		0.3–1
	[48]		0.31
Mars surface	[19]		0.17–0.325
	[49]		0.002–0.9
	[52]		0.15
	[51]		0.33

**Table 2 ijms-22-03739-t002:** Current average effective dose and dose-rates of interest for different exposures to IR.

		On Earth	In Space
**Effective dose** **(mSv)**	700–12,000	Acute radiation syndrome (irradiation accident)	ARS during solar events?
	2000	Tumor dose per fraction in radiotherapy Dose per fraction in total body irradiation	
	200	Threshold dose for solid cancer	500 days in ISS
	100	Threshold dose for leukemia	
	20	Coronarography	50 days in ISS
	0.3–0.5	2 mammography views	1 day in ISS
	0.01	1 day for a nuclear worker	
**Effective dose-rate (mSv/y)**	0.5	Radiation background in Japan	
	2.4	Worldwide radiation background	
	70	Radiation background in Ramsar	
	110–180		In LEO (ISS)
	130–260		At the surface of Mars
	110–300		At the surface of the Moon
	500–700		In deep space

**Table 3 ijms-22-03739-t003:** Major radiation-induced risks for astronauts after long-term space mission in absence of severe solar events.

Radiation-Induced Consequences	Type of Space Radiation	Type of Tissues	Clinical Consequences	Countermeasures
**Radiosusceptibility** The 100–200 mGy thresholds for cancer risks derived from Hiroshima survivors are the only consensual series of data.	Low energy ions/particles including neutrons	*“surface tissues”:* Eye Skin	Eye melanoma Skin melanoma	Glasses? Water-filled worksuits?
	“Bath of radiation” including fast neutrons	*“deep tissues”:* Cardiovascular System Bones General	No cancer observed yet Osteosarcoma Leukemia	Radioprotective drugs
**Radiodegeneration** There are no well-defined dose thresholds (>0.1 Gy?) nor biomarkers that are specific to aging yet.	Low energy ions/particles including neutrons	*“surface tissues”:* Eye Skin	Cataracts Aging	Glasses? Water-filled worksuits?
	“Bath of radiation” including fast neutrons	*“deep tissues”:* Cardiovascular System Bones General	Heart attacks Loss of bone mass To be investigated	Radioprotective drugs

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
