# Peer review of "Radiation on Earth or in Space: What Does It Change?"

_ijms, 2021, doi:10.3390/ijms22073739_

Round 1

Reviewer 1 Report

Manuscript Number: IJMS- 1140584

Title: Radiation on Earth or in Space: What does it change?

Recommendation: Reject

The manuscript tries to summarize the knowledge on radiation exposure of astronauts, resulting health risks and suggests countermeasures. The description of the composition of radiation field inside spacecrafts and inside the human body and of the effects of shielding reveals several misconceptions which are detailed below.  

A major point in this review is the conlusion that low energy particles produce a relatively high dose to the eyes and the skin inside a spacecraft. Such a high skin dose was found during extravehicular activity in a spacesuit but is strongly ameliorated inside the ISS (see MATROSHKA phantom measurements on ISS).

Furthermore, the manuscript is hampered by the missing of (citation of) dose measurements on ISS, in free space and on the surface of Moon and Mars.

Also, the description of health risks is incomplete and mixed with terms of individual radiosensitivity.

In conclusion, this manuscript produces confusion by ignoring existing data and should be rejected.

Abstract

“space particle” – this is a rather unusual term for high energy charged particles present in cosmic radiation. It could cause confusion with space debris.

“their contribution to the dose very low” => do you mean energy dose or effective dose?

“A “bath” of radiation of 146 mSv/y” => the dose accumulated in low Earth orbit depends on many parameters such as solar activity, shielding of the spacecraft, flight height, orbit and traversal of the South Atlantic Anomaly. Therefore, it would be better to indicate a dose range per year rather than a discrete dose per year. Also, please explain what you mean by “bath”, a chronic whole-body exposure?

It is unclear why especially bones are highlighted here – because of the bone marrow?

Skin melanoma: it is unclear whether the non-significant increase in the US astronaut population is related to cosmic ray exposure.

Introduction

  1. 2 “the existence of the Earth’s radiation belt that acts as a magnetic shield against charged particles, the Van Allen radiation belt” => the original work rather says it like this: “a high intensity of corpuscular radiation temporarily trapped in the Earth's magnetic field”. It does not say that the radiation belt acts as magnetic shield.

  1. The different sources of ionizing radiation in space

2.1. Discovery of cosmic rays, Van Allen belt and South Atlantic anomaly

  1. 3: solar flashes => solar flares

2.2. The energy-distribution and the impact probability

  1. 3: 14% helium ions (α particles) => it is recommended not to call the helium ions of GCR alpha-particles to avoid confusion with alpha-particles from a radioactive decay which have much lower energy.

  1. 3: “the probability of impact by heavy ions to astronauts can be therefore considered as negligible” => such a statement should be accompanied by an estimation of heavy ion hits per body cell for an exploration mission such as a Mars mission.

  1. 3: “Even if the space missions are scheduled to avoid such events, some missions like Apollo-XIV, Skylab-4, and Mir-15 have been submitted to some significant solar events, at least partially: the crews received 11.4 mGy in 9 days, 77 mGy in 90 days, and 92.9 mGy in 185 days, respectively (Figure 2)”

=> Please specify whether this exposure was in addition to the GCR exposure or included the GCR exposure.

  1. 3: “In geostationary orbit” => Mir was not located in geostationary orbit (~36 000 km above Earth’s surface), but in low Earth orbit (354-374 km)

  1. 4: it is unclear how the formula Foutside was derived. Can you please cite the source or explain how it was developed? Usually, radiation transport codes are used.

  1. 4: Please explain why the high energy helium ions are stopped by spacecraft shielding and cite relevant measurements.

2.3. The space radiation that actually concern astronauts - Controversies

  1. 4: “The effective dose equivalent was recently measured to be 3 mSv/d outside the International Space Station (ISS) and 0.6 mSv/d inside the ISS to the skin” => an effective dose cannot be measured but only calculated. Phantom experiments help in determination of the organ doses that are required for the calculation.

  1. 4-5: Description of effects of shielding: references to actual measurements behind shielding of different material and thickness are missing. Neutrons were identified as important secondary particles, but they are not mentioned at all in this section.

  1. 5: “GCR particles show a very low probability of impact to astronauts in their spacecraft” => do you mean the heavy ion component alone or also protons and helium ions present in GCR?

Figure 2 Legend

  1. 6 “The dark bar represents the average radiation dose rate inside spacecraft (146 mSv/y) around the Earth, between the Earth and the Moon.”

The bar extends also to Mars. From which radiation measurements do you derive the conclusion that the dose rate inside a spacecraft is the same in low Earth orbit and in free space on the way to Moon and Mars?

Figure 2, right panel: Please cite the source of the radiation measurements in low Earth orbit, and on Moon and Mars.

2.4 Comparison with the natural radiation backgrounds on the Earth, Moon and Mars

  1. 6 “A dose-rate of about 0.3 mSv/y comes from the organic products that are naturally radioactive like potassium (40K) and carbon (14C)” => “A dose-rate of about 0.3 mSv/y comes from the organic products that contain naturally radioactive like potassium (40K) and carbon (14C)”

  1. 6 “Besides, it is noteworthy that there is no significant difference in cancer occurrence rate between Japan and Ramsar” => please cite the relevant epidemiological surveys.

  1. 7 “Indeed, the surface of the Moon and Mars are much more radioactive that that of Earth: according to the moment of the day, the natural radiation backgrounds assessed on the Moon and Mars are 110-380 and 130-260 mSv/y, respectively.” => Does this refer to radioactive decays in the rocks or to GCR and secondary radiation?

  1. The potential radiation-induced risks for the astronauts: what do we expect?

  1. 7 “IR can produce three distinct types of clinical effects : radiosensitivity, radiosusceptibility and radiodegeneration” => radiosensitivity and radiosusceptibility are inherent characteristics of an individuum and are not produced by radiation.

  1. 8 “However, it must be stressed that neutrons, LEMI, LEP and LEE may also concern aircraft pilots, like astronauts.” => Modeling revealed only a very small difference between skin dose and effective dose for the exposure of aircrew to cosmic radiation (maximum factor 1.09):
  2. Radiol. Prot. 37 (2017) 321–328 (8pp) https://doi.org/10.1088/1361-6498/aa5eef

  1. 8 “An increased risk of cataract was generally observed after exposure to doses above 8 mGy, which corresponds to approximately 20 mission days.” – Please cite the original publication. The most important difference between the two astronaut groups (< 8 mGy / > 8 mGy) was the higher inclination orbit for the “high-dose” group.

  1. 9: “However, while the molecular and cellular pathways by which microgravity may act into biochemical signals are still unknown,” => In microgravity, weight-bearing parts of the musculoskeletal system are unloaded, resulting in bone remodeling that adapts bone strength to the mechanical strains prevailing during a space mission. A “direct effect of microgravity on cells” is not necessary to explain the bone loss.

  1. 9: “Epidemiology data of breast cancer women patients have shown that more than 50% of women are at risk of heart attack for 10 years post-radiotherapy [43-46]. Such examples, suggest that cardiovascular system may be mechanically affected by radiation and sensitive to low dose.” => Why “mechanically” affected? Why do you conclude that the heart of the breast cancer patients was exposed to a “low dose”? What do you consider as a low dose?

Table 1:

Radiation-induced Consequences: radiosusceptibility is an inherent characteristic of an individuum and is not produced by radiation.

  1. Countermeasures and actions to better estimate the radiation-induced risks for astronauts

  1. 11: “To reduce the specific risks linked to the low-energy particles, a simple protection may be sufficient: indeed, special glasses and worksuits may protect both skin and eye significantly.” => please cite the relevant measurements of low-energy particles inside the ISS supporting this suggestion. The water-filled garment is intended for additional protection during solar particle events.

  1. 12: “WR radiation weighting factor” => for operational purposes in space missions, the quality factor, Q(L) can be used. NASA uses the quality factor (QF) that depends on Z2/beta2 in its radiation risk model.

  1. 12: “loss of bone mass and accelerated of cardiovascular system)” => accelerated ageing?

Sufficient evidence to attribute bone loss to GCR exposure is lacking. The low dose exposure might contribute only to a very minor part to it.

Cardiovascular system: the current conception is that the risks to the central nervous system are higher than to the cardiovascular system.

Figures:

A reference to Fig. 1 is missing in the introduction.

Fig. 1 and its legend:

Galactic component radiation (GCR) => Galactic cosmic radiation (or rays) (GCR)

protontherapy, electrontherapy => proton therapy, electron therapy (check the whole manuscript)

CT => please explain all abbreviations at their first appearance in the abstract, the main text or figure legends

Bath of radiation: 146 mSv/y” => see comments on abstract

Minor points:

The manuscript needs a thorough language check by a native speaker. Below are only some examples for necessary corrections.

  1. 1: IR remove => IR removes

  1. 1: to be more documented => to be documented better

  1. 3: “sun” => “Sun”

Author Response

REVIEWER 1

We thank the reviewer for his/her comments. It is noteworthy that, to reach all the requirements of the 3 reviewers, the text has been deeply modified, amplified and more than 40 new references have been added. However, since some comments may have been contradictory among the reviewers, some specific choices have been made in the corrected version.

The manuscript tries to summarize the knowledge on radiation exposure of astronauts, resulting health risks and suggests countermeasures. The description of the composition of radiation field inside spacecrafts and inside the human body and of the effects of shielding reveals several misconceptions which are detailed below.  

A major point in this review is the conlusion that low energy particles produce a relatively high dose to the eyes and the skin inside a spacecraft. Such a high skin dose was found during extravehicular activity in a spacesuit but is strongly ameliorated inside the ISS (see MATROSHKA phantom measurements on ISS).

  1. We agree that high skin dose is strongly ameliorated inside the ISS by water-filled garments suits (newly cited reference) but conversely low energy particles may be important during the journey to Moon or to Mars  that’s why we have cited in the first version the paper about Apollo’s helmets. Furthermore, it is difficult to conciliate two reviewers’comments that are contradictory: we have modified a paragraph focusing on low energy particles that may be important and documented it more notably to reply to the reviewer 3. The papers about MATROSHKA phantom have been also cited. See modified text page 4 para 2.3, page 6 and page 16

Furthermore, the manuscript is hampered by the missing of (citation of) dose measurements on ISS, in free space and on the surface of Moon and Mars.

  1. We have added an important amount of values concerning Moon and Mars on one side and on ISS on the other side. See modified text pages 4 to 6 and new tables 1 and 2

Also, the description of health risks is incomplete and mixed with terms of individual radiosensitivity.

Individual radiosensitivity is one of the three notions of priority for NASA and ESA but also of EURATOM and EC research institutions. Since the simple application of the Sievert system is not consensual among the clinicians, we have chosen to survey all the potential radiation-induced pathologies by dispatching them in the three categories radiosensitivity/radiosusceptibility/radiodegenerescence. It is important to note that ICRP as a new group about individual radiosensitivity that has adopted these terms. As replied to the reviewer 2, we were also obliged to present the risks of radiodegenerescence as potential risks since there is still neither consensual thresholds nor specific biomarkers. Thereafter, we became aware that a number of organs considered by NASA at risk were derived from the ARS syndrome, only relevant during extreme solar event. Because of all these uncertainties, there were few numbers in the last table. We have therefore deeply modified the text to better illustrate the overall health outcomes. See modified text page 6 para 2.5, new table 2 , page 10 para 3.1, beginning of the page 11, pages 13 and 14.

In conclusion, this manuscript produces confusion by ignoring existing data and should be rejected.

 The text has been deeply modified, amplified and more than 40 new references have been added.

Abstract

“space particle” – this is a rather unusual term for high energy charged particles present in cosmic radiation. It could cause confusion with space debris.

  1. You are right but if we write cosmic particle, it means particle emitted from cosmos and not necessarily Sun or Van Allen belt. We endeavoured to be more precise in the terms used, with the risk to avoid usual jargon. We proposed “radiation emitted from space” knowing that radiation is used to designed both rays and particles. See modified text page1 Introduction line 6.

“their contribution to the dose very low” => do you mean energy dose or effective dose?

OK First, to the absorbed dose and thereafter, after calculation to the effective dose. See modified text in abstract and new table 1

“A “bath” of radiation of 146 mSv/y” => the dose accumulated in low Earth orbit depends on many parameters such as solar activity, shielding of the spacecraft, flight height, orbit and traversal of the South Atlantic Anomaly. Therefore, it would be better to indicate a dose range per year rather than a discrete dose per year.

OK see modified text in all the paper. However, to reach another reviewer requirement, we have expressed data in mSv/d as well.

Also, please explain what you mean by “bath”, a chronic whole-body exposure?

Yes. See modified text pages 5 and 6

It is unclear why especially bones are highlighted here – because of the bone marrow?

Yes. For both bone marrow but overall for IRS-induced loss of bone that a major and systematic clinical feature of astronauts. See modified text in Abstract and see again the end of page 12.

Skin melanoma: it is unclear whether the non-significant increase in the US astronaut population is related to cosmic ray exposure.

  1. See modified text page 11.

Introduction

  1. 2 “the existence of the Earth’s radiation belt that acts as a magnetic shield against charged particles, the Van Allen radiation belt” => the original work rather says it like this: “a high intensity of corpuscular radiation temporarily trapped in the Earth's magnetic field”. It does not say that the radiation belt acts as magnetic shield.

 OK see modified text page 2 Para 2.1

  1. The different sources of ionizing radiation in space

2.1. Discovery of cosmic rays, Van Allen belt and South Atlantic anomaly

  1. 3: solar flashes => solar flares

 OK see modified text page 2 Para 2.1 and Page 3 para 2.2

2.2. The energy-distribution and the impact probability

  1. 3: 14% helium ions (α particles) => it is recommended not to call the helium ions of GCR alpha-particles to avoid confusion with alpha-particles from a radioactive decay which have much lower energy.

 OK see modified text page 3 para 2.2

  1. 3: “the probability of impact by heavy ions to astronauts can be therefore considered as negligible” => such a statement should be accompanied by an estimation of heavy ion hits per body cell for an exploration mission such as a Mars mission.
  1. This part of the text has been deleted and modified. See modified text page 3 para 2.2

  1. 3: “Even if the space missions are scheduled to avoid such events, some missions like Apollo-XIV, Skylab-4, and Mir-15 have been submitted to some significant solar events, at least partially: the crews received 11.4 mGy in 9 days, 77 mGy in 90 days, and 92.9 mGy in 185 days, respectively (Figure 2)”

=> Please specify whether this exposure was in addition to the GCR exposure or included the GCR exposure.

  1. See modified text last lines of page 3

  1. 3: “In geostationary orbit” => Mir was not located in geostationary orbit (~36 000 km above Earth’s surface), but in low Earth orbit (354-374 km)
  1. You are right. see modified text page 4 first para

  1. rgus 4: it is unclear how the formula Foutside was derived. Can you please cite the source or explain how it was developed? Usually, radiation transport codes are used.
  1.  Such formula is an empirical one that fits well the proton component. Ii is not contradictory with the use of radiation transport codes. It will serve for other developments. See modified text page 4 formula 1

  1. 4: Please explain why the high energy helium ions are stopped by spacecraft shielding and cite relevant measurements.
  1. Text has been rephrased. See modified text page 3 para 2.2

2.3. The space radiation that actually concern astronauts - Controversies

  1. 4: “The effective dose equivalent was recently measured to be 3 mSv/d outside the International Space Station (ISS) and 0.6 mSv/d inside the ISS to the skin” => an effective dose cannot be measured but only calculated. Phantom experiments help in determination of the organ doses that are required for the calculation.
  1. This sentence has been deleted See modified text page 4.

  1. 4-5: Description of effects of shielding: references to actual measurements behind shielding of different material and thickness are missing.

 OK see modified text page 4 para 2.3 and page 5

 Neutrons were identified as important secondary particles, but they are not mentioned at all in this section.

 OK see modified text page 5 before para 2.4, and page 6 before para 2.5.

  1. 5: “GCR particles show a very low probability of impact to astronauts in their spacecraft” => do you mean the heavy ion component alone or also protons and helium ions present in GCR?

Yes the heaviest ions :  see modified text page 3 para 2.2 and page 6 before para 2.5

Figure 2 Legend

  1. 6 “The dark bar represents the average radiation dose rate inside spacecraft (146 mSv/y) around the Earth, between the Earth and the Moon.”

The bar extends also to Mars. From which radiation measurements do you derive the conclusion that the dose rate inside a spacecraft is the same in low Earth orbit and in free space on the way to Moon and Mars?

OK We did not say that but you are right, it may create a misunderstanding :see modified Figure and legend of Figure 2 and new table 1

Figure 2, right panel: Please cite the source of the radiation measurements in low Earth orbit, and on Moon and Mars.

  OK see modified text and new table 1

2.4 Comparison with the natural radiation backgrounds on the Earth, Moon and Mars

  1. 6 “A dose-rate of about 0.3 mSv/y comes from the organic products that are naturally radioactive like potassium (40K) and carbon (14C)” => “A dose-rate of about 0.3 mSv/y comes from the organic products that contain naturally radioactive like potassium (40K) and carbon (14C)”

 OK see modified text page para 2.6

  1. 6 “Besides, it is noteworthy that there is no significant difference in cancer occurrence rate between Japan and Ramsar” => please cite the relevant epidemiological surveys.

 OK see modified text page 8 para 2.6

  1. 7 “Indeed, the surface of the Moon and Mars are much more radioactive that that of Earth: according to the moment of the day, the natural radiation backgrounds assessed on the Moon and Mars are 110-380 and 130-260 mSv/y, respectively.” => Does this refer to radioactive decays in the rocks or to GCR and secondary radiation?

OK . The measurements included both. See modified text  and cited references and table 1

  1. The potential radiation-induced risks for the astronauts: what do we expect?
  1. 7 “IR can produce three distinct types of clinical effects : radiosensitivity, radiosusceptibility and radiodegeneration” => radiosensitivity and radiosusceptibility are inherent characteristics of an individuum and are not produced by radiation.
  1. there is a large body evidence that cancer proneness is dependent on the dose, on the genetic status and both. For example, BRCA1- or p53- mutated patients are at spontaneous and radiation-induced risk of cancer. Similarly, radiosensitivity is dependent on dose and/or genetic status. In other terms, the dose-effects or risk function of dose slope change with mutations. See pages 12,13

8 “However, it must be stressed that neutrons, LEMI, LEP and LEE may also concern aircraft pilots, like astronauts.” => Modeling revealed only a very small difference between skin dose and effective dose for the exposure of aircrew to cosmic radiation (maximum factor 1.09):

Radiol. Prot. 37 (2017) 321–328 (8pp) https://doi.org/10.1088/1361-6498/aa5ee

OK we cited the paper but there is nothing in this paper that may evoke the importance of neutrons, LEMI, MEP and LEE outside the LEO. Furthermore, the current Wr factor of LEMI, LEP and LEE is not consensual at all and the reviewer 3 agrees this hypothesis that may lead to the modulation of the conclusions of the authors of this paper. See modified text page 11 and 5.

8 “An increased risk of cataract was generally observed after exposure to doses above 8 mGy, which corresponds to approximately 20 mission days.” – Please cite the original publication. The most important difference between the two astronaut groups (< 8 mGy / > 8 mGy) was the higher inclination orbit for the “high-dose” group.

OK see references page 12 para 3.3

9: “However, while the molecular and cellular pathways by which microgravity may act into biochemical signals are still unknown,” => In microgravity, weight-bearing parts of the musculoskeletal system are unloaded, resulting in bone remodeling that adapts bone strength to the mechanical strains prevailing during a space mission. A “direct effect of microgravity on cells” is not necessary to explain the bone loss.

Yes that’s why we have insisted on the radiosensitivity of bone, a new feature to be considered since the microgravity does not explain all the loss of bone, notably Fairlet et al 202 supported by our own data to be published. See text page 13.

  1. 9: “Epidemiology data of breast cancer women patients have shown that more than 50% of women are at risk of heart attack for 10 years post-radiotherapy [43-46]. Such examples, suggest that cardiovascular system may be mechanically affected by radiation and sensitive to low dose.” => Why “mechanically” affected? Why do you conclude that the heart of the breast cancer patients was exposed to a “low dose”? What do you consider as a low dose?

Mechanically by the fact that aged cardiac tissue loses its properties. During a radiotherapy of the brest, apex may received mGy.

Table 1:

Radiation-induced Consequences: radiosusceptibility is an inherent characteristic of an individuum and is not produced by radiation.

Again, the reviewer must consider the risk of spontaneous and the radiation-induced cancer and the interplay with the genetic cancer susceptibility. Again, the risk curve as a function of dose is clearly different when we consider BRCA1, p53, ATM carriers.

  1. Countermeasures and actions to better estimate the radiation-induced risks for astronauts
  1. 11: “To reduce the specific risks linked to the low-energy particles, a simple protection may be sufficient: indeed, special glasses and worksuits may protect both skin and eye significantly.” => please cite the relevant measurements of low-energy particles inside the Cardiovascular system: the current conception is that the risks to the central nervous system are higher than to the cardiovascular system. ISS supporting this suggestion. The water-filled garment is intended for additional protection during solar particle events.

The current conceptions about the central nervous system have been suggested from works about the acute radiation syndromes (i.e. scenario linked to solar events). The questions raised by cardiovascular system are raised by low doses of radiation as mentioned. The radiobiological features of brain and heart are clearly different. See modified text page 10 para 3.1

On another hand, we have cited the works about the water-filled garment suits but the questions about low energy particules remains in the deep space. See modified text page 16 para 5.

  1. 12: “WR radiation weighting factor” => for operational purposes in space missions, the quality factor, Q(L) can be used. NASA uses the quality factor (QF) that depends on Z2/beta2 in its radiation risk model.

Yes but QF is not consensual and the reviewer 3’s comments must be addressed. Furthermore,   it is noteworthy that WR depends also on RBE and RBE of low-energy particle sand even X-rays cannot be 1. 

  1. 12: “loss of bone mass and accelerated of cardiovascular system)” => accelerated ageing?

Yes see modified text page 17

Sufficient evidence to attribute bone loss to GCR exposure is lacking. The low dose exposure might contribute only to a very minor part to it.

 No if you consider that bone is a radiosensitive tissue, notably radiosensitive to low dose as the Failey et aL; 2020 paper suggest it. Our recent findings are in agreement with them

Figures:

A reference to Fig. 1 is missing in the introduction.

 OK see modified Fig.1

Fig. 1 and its legend:

Galactic component radiation (GCR) => Galactic cosmic radiation (or rays) (GCR)

OK see modified text page 2 Para 2.1

protontherapy, electrontherapy => proton therapy, electron therapy (check the whole manuscript)

OK see modified text in the legend of figure 1 and in para 2.1

CT => please explain all abbreviations at their first appearance in the abstract, the main text or figure legends

OK see modified text in the legend of figure 1

Bath of radiation: 146 mSv/y” => see comments on abstract. OK see modified text

Minor points:

The manuscript needs a thorough language check by a native speaker. Below are only some examples for necessary corrections.

  1. 1: IR remove => IR removes

 “Radiation are” like “damage are” is allowed, since there terms represents both countable and non countable forms :  that’s why “IR remove” is correct

  1. 1: to be more documented => to be documented better

 OK see modified text page 1 Introduction line 8

  1. 3: “sun” => “Sun”

  OK see modified text

Reviewer 2 Report

This is an interesting review about an important issue. I would like the authors to address internal exposures to proton and neutron activation of gases and materials generatied in the spacecraft. This is missing from their analysis. One might expect a build up of gamma emitters and other activation products in the spacecraft material. There is no mention of the effects of long term exposures of the spacecraft material to cosmic rays. The discussion of the health effects is weak, and is based on the authors own ideas and definitions of radiation effects on health and an acceptance of the Japanese A-Bomb studies dose response effects for cancer. Thus  their conclusions about the overall health outcomes of exposures are questionable. I would like the authors to address these points. In general they are trying to cram too many things into one paper. I would not recommend publication until these issues are addressed. However I think they could do this, and they could be a bit more cautious about the levels of dose they think represent hazards. I agree with their suggestion that there should be an ICRP external  dose quality factor based on energy, since low energy gamma and x-rays will produce clustered ionisation events.

Author Response

REVIEWER 2

We thank the reviewer for his/her comments. It is noteworthy that, to reach all the requirements of the 3 reviewers, the text has been deeply modified, amplified and more than 40 new references have been added. However, since some comments may have been contradictory among the reviewers, some specific choices have been made in the corrected version

Comments and Suggestions for Authors

This is an interesting review about an important issue. I would like the authors to address internal exposures to proton and neutron activation of gases and materials generatied in the spacecraft. This is missing from their analysis. One might expect a build up of gamma emitters and other activation products in the spacecraft material.

  1. Yes you are right and we fully agree: however, it was difficult to find some reports about activation. We have therefore introduced an entire paragraph about the neutron activation potentially occurring in the spacecraft (see modified text page 4 para 2.3 and page 5)

There is no mention of the effects of long term exposures of the spacecraft material to cosmic rays.

We have focused on health issues and considered the aging effects on material as not in the scope of the review.

The discussion of the health effects is weak, and is based on the authors own ideas and definitions of radiation effects on health and an acceptance of the Japanese A-Bomb studies dose response effects for cancer. Thus, their conclusions about the overall health outcomes of exposures are questionable. I would like the authors to address these points.

  1. As a first step, we were obliged to present first the general point of view about health issues for astronauts based on the current paradigms and radiation protection regulations and notably on the only consensual data provided by the Hiroshima survivors data (cancer risks thresholds of 100-200 mGy). Secondly, we were also obliged to present the risks of radiodegenerescence as potential risks since there is still neither consensual thresholds nor specific biomarkers. Thirdly, we became aware that a number of organs considered by NASA at risk were derived from the ARS syndrome, only relevant during extreme solar event. Because of all these uncertainties, there were few numbers in the last table. We have therefore deeply modified the text to better illustrate the overall health outcomes. See modified text page 6 para 2.5, new table 2 , page 10 para 3.1, beginning of the page 11, pages 13 and 14.

In general, they are trying to cram too many things into one paper. I would not recommend publication until these issues are addressed. However I think they could do this, and they could be a bit more cautious about the levels of dose they think represent hazards.

OK we endeavoured to present an overview of the quantification of hazards to provide the most probable thresholds and levels of risks. See modified text page 6 para 2.5, new table 2 , page 10 para 3.1, beginning of the page 11, pages 13 and 14.

I agree with their suggestion that there should be an ICRP external dose quality factor based on energy, since low energy gamma and x-rays will produce clustered ionisation events.

OK; We have developed again this subject. See modified text page 13 para 4.1

Reviewer 3 Report

This manuscript describe a qualitative review of the risks linked to exposure to space radiation.

The manuscript describes also a qualitative list of the expected effects of space radiation and possible countermeasures.
What emerges form this manuscript is the lacks of quantitative predictions due to missing data collected in space.
This is evident also reading table 1 that is containing:
20 cells, 8 question marks and any numbers.

Being the title of the manuscript:
"Radiation on Earth or in Space: What does it change?"
and a lot of data exists for radiation effects in Earth, I would expect to read in this review some quantitative comparison and prediction of effects for space with the relative (maybe large) uncertainty due to extrapolation from ground data.

Here some suggested improvements:
1) In equation 2 a symbol \otimes is missing before S(E)

2) Fig. 2 please increase the size
I cannot find Footnote "missions^2"

3) in many parts of the text (and in fig.1) is indicated 146mSv/y
suggesting an optimistic uncertainty below 1% for this number.
Please replace everywhere with 0.4mSv/d

4) Pag 4 title of section 2.3
"The space radiation that actually concern astronauts - Controversies"
In the text are not well stated the "controversies"

5) Pag 5 last sentence
"This picture is qualitatively different
from that described in the 1960" please add a reference
to the old picture

6) pag 7
"according to the moment of the day, the natural radiation
backgrounds assessed on the Moon and Mars are 110-380
and 130-260 mSv/y, respectively"
Please complete the sentence adding some plausible explanation or
cite a reference containing this explanation.

Author Response

REVIEWER 3

We thank the reviewer for his/her comments. It is noteworthy that, to reach all the requirements of the 3 reviewers, the text has been deeply modified, amplified and more than 40 new references have been added. However, since some comments may have been contradictory among the reviewers, some specific choices have been made in the corrected version

Comments and Suggestions for Authors

This manuscript describe a qualitative review of the risks linked to exposure to space radiation. The manuscript describes also a qualitative list of the expected effects of space radiation and possible countermeasures.What emerges form this manuscript is the lacks of quantitative predictions due to missing data collected in space. This is evident also reading table 1 that is containing:
20 cells, 8 question marks and any numbers.

  1. You are right. We have added more than 40 references and quantitative values at each step of the review in order to be more precise and provide a quantitative view of risks. As also replied to the reviewer 2, as a first step, we were obliged to present first the general point of view about health issues for astronauts based on the current paradigms and radiation protection regulations and notably on the only consensual data provided by the Hiroshima survivors data (cancer risks thresholds of 100-200 mGy). Secondly, we were also obliged to present the risks of radiodegenerescence as potential risks since there is still neither consensual thresholds nor specific biomarkers. Thirdly, we became aware that a number of organs considered by NASA at risk were derived from the ARS syndrome, only relevant during extreme solar event. Because of all these uncertainties, there were few numbers in the last table. We have therefore deeply modified the text to better illustrate the overall health outcomes. See modified text page 6 para 2.5, new table 2 , page 10 para 3.1, beginning of the page 11, pages 13 and 14.

Being the title of the manuscript:"Radiation on Earth or in Space: What does it change?" and a lot of data exists for radiation effects in Earth, I would expect to read in this review some quantitative comparison and prediction of effects for space with the relative (maybe large) uncertainty due to extrapolation from ground data.

OK . You are right. Particularly, we have added a new Table 2 for quantitative comparisons.

Here some suggested improvements:
1) In equation 2 a symbol \otimes is missing before S(E)

  1. To clarify the text, we have deleted this equation. See modified text

2) Fig. 2 please increase the size
I cannot find Footnote "missions^2"

OK See modified legend to figure. The 2 of mission2 was a typo.

3) in many parts of the text (and in fig.1) is indicated 146mSv/y suggesting an optimistic uncertainty below 1% for this number. Please replace everywhere with 0.4mSv/d

OK you are right. See modified text. However, since radiation backgrounds are generally compared when expressed in mSv/y, some sentences and values may have been kept with this unit. See modified text all along the text.

4) Pag 4 title of section 2.3
"The space radiation that actually concern astronauts - Controversies"
In the text are not well stated the "controversies"

  1. You are right. Title has been changed and the term “controversies” has been deleted. See modified text.

5) Pag 5 last sentence
"This picture is qualitatively different from that described in the 1960" please add a reference
to the old picture

  1. This sentence has been deleted.

6) pag 7
"according to the moment of the day, the natural radiation
backgrounds assessed on the Moon and Mars are 110-380
and 130-260 mSv/y, respectively"

Please complete the sentence adding some plausible explanation or
cite a reference containing this explanation.

OK See modified text page 6 before para 2.5

Round 2

Reviewer 3 Report

Dear author

there are still few improvement that should be included in the manuscript:

Question1: pag5, the sentence:
"To activate water or oxygen inside spacecraft is quite difficult since the 19 O oxygen is very unstable."
is wrong.
The lifetime of a daughter isotope is not related to the fragmentation cross section or to neutron capture cross section of the parent nuclei. Therefore to activate oxygen is not difficult because of the short 19O lifetime. Most probably aim of the author, including this sentence, is to point out that due to the short lifetime of oxygen isotopes they unlikely are found in the body of the astronauts (despite they are indeed produced in the spacecraft and within the astronaut body)
However beyond 19O (T1/2=26s) I would mention also 15O (T1/2=122s) that is produced by neutron photo-emission or spallation.

Question2: pag8, the given numbers are not consistent each others:
telluric = 1.2mSv/y
organic = 0.3mSv/y
cosmic = 0.3mSv/y but it is written that is only 0.2% of the previous ones.
average worldwide = 2.4mSv/y that is larger than (1.2+0.3+0.3)mSv/y
(therefore the previous one was not averaged but specific of some site)
No reference is given for these numbers.

Please try to give numbers that are in agreement each other.

Question 3: pag.8
If possible try to quantify these sentences summarizing the cancer occurrence probability ratio and uncertainty(form the text I would expect that the occurrence probability difference of first report is not compatible with 140 and that the second ratio is not compatible with 1 and is below 1):  

"It is noteworthy that there is no a 140-fold
difference in cancer occurrence rate between Japan and Ramsar"
and
"By contrast, some reports have suggested a lower risk of cancer and radiation-
induced diseases in Ramsar (hormesis phenomenon)."

Question 4: pag 14:
"the dose-response curve non-linearly"
This review has a lot of qualitative descriptions and some
tables but the interesting "dose-response" plot is missing.

A plot with "measured disease probability" vs "measured dose" including measurement error bars would be the plot I expect to see in this review. It would be a useful and quantitative summary of existing data. 

Author Response

REVIEWER’s COMMENTS

We thank the Reviewer for his/her comments

there are still few improvement that should be included in the manuscript:

Question1: pag5, the sentence:
"To activate water or oxygen inside spacecraft is quite difficult since the 19 O oxygen is very unstable."
is wrong.
The lifetime of a daughter isotope is not related to the fragmentation cross section or to neutron capture cross section of the parent nuclei. Therefore to activate oxygen is not difficult because of the short 19O lifetime. Most probably aim of the author, including this sentence, is to point out that due to the short lifetime of oxygen isotopes they unlikely are found in the body of the astronauts (despite they are indeed produced in the spacecraft and within the astronaut body)
However beyond 19O (T1/2=26s) I would mention also 15O (T1/2=122s) that is produced by neutron
photo-emission or spallation.

OK see modified text page 5

Question2: pag8, the given numbers are not consistent each others:
telluric = 1.2mSv/y
organic = 0.3mSv/y
cosmic = 0.3mSv/y but it is written that is only 0.2% of the previous ones.
average worldwide = 2.4mSv/y that is larger than (1.2+0.3+0.3)mSv/y
(therefore the previous one was not averaged but specific of some site)
No reference is given for these numbers

Please try to give numbers that are in agreement each other.

OK see modified text page 8

Question 3: pag.8
If possible try to quantify these sentences summarizing the cancer occurrence probability ratio and uncertainty(form the text I would expect that the occurrence probability difference of first report is not compatible with 140 and that the second ratio is not compatible with 1 and is below 1):  

"It is noteworthy that there is no a 140-fold
difference in cancer occurrence rate between Japan and Ramsar"
and
"By contrast, some reports have suggested a lower risk of cancer and radiation-
induced diseases in Ramsar (hormesis phenomenon)."

Your comment seems to be unclear but we have modified the text page 8

Question 4: pag 14:
"the dose-response curve non-linearly"
This review has a lot of qualitative descriptions and some
tables but the interesting "dose-response" plot is missing.

A plot with "measured disease probability" vs "measured dose" including measurement error bars would be the plot I expect to see in this review. It would be a useful and quantitative summary of existing data. 

You are fully right but unfortunately the quantitative dose-response curves specific to each radiation-induced diseases (cataracts, eye and skin melanoma, loss of bone, aging of cardiovascular system ) are still unknown because of non-specific biomarkers. The picture you ask is inasmuch difficult to draw as the dose-rate should be the most relevant x-axis criteria  : such dose-rate -responses expected are still more difficult to obtain to date, considering the lack of quantitative data in literature.